# Identification of miRNA–mRNA Networks Associated with Pigeon Skeletal Muscle Development and Growth

**DOI:** 10.3390/ani12192509

**Published:** 2022-09-21

**Authors:** Hao Ding, Can Chen, Tao Zhang, Lan Chen, Weilin Chen, Xuanze Ling, Genxi Zhang, Jinyu Wang, Kaizhou Xie, Guojun Dai

**Affiliations:** 1College of Veterinary Medicine, Yangzhou University, Yangzhou 225000, China; 2Joint International Research Laboratory of Agriculture and Agri-Product Safety, Ministry of Education, Yangzhou University, Yangzhou 225000, China; 3College of Animal Science and Technology, Yangzhou University, Yangzhou 225000, China

**Keywords:** pigeon, skeletal muscle, miRNA–mRNA network, RNA sequencing

## Abstract

**Simple Summary:**

Elucidating the molecular mechanisms that regulate the growth and development of skeletal muscle is an essential prerequisite for using molecular breeding technology to improve the meat production of pigeons. In the present study, we characterized the expression of miRNAs and mRNAs in pigeon skeletal muscle at two embryonic stages (E8 and E13) and two growth stages (D1 and D10). A total of 839 DEmiRNAs and 11,311 DEGs were identified among the four groups. An miRNA–mRNA network was constructed by correlation analysis and target prediction between DEmiRNAs and DEGs. Based on the connectivity degree, twenty potential hub miRNAs responsible for pigeon skeletal muscle development and growth were identified, including cli-miR-20b-5p, miR-130-y, cli-miR-106-5p, cli-miR-181b-5p, miR-1-z, etc. GO and KEGG enrichment analysis identified candidate biological processes and pathways related to pigeon skeletal muscle development and growth. Our results provide a database for further investigating the miRNA–mRNA regulatory mechanism underlying pigeon skeletal muscle development and growth.

**Abstract:**

The growth and development of skeletal muscle determine the productivity of pigeon meat production, and miRNA plays an important role in the growth and development of this type of muscle. However, there are few reports regarding miRNA regulating the growth and development of skeletal muscle in pigeons. To explore the function of miRNA in regulating the growth and development of pigeon skeletal muscle, we used RNA sequencing technology to study the transcriptome of pigeons at two embryonic stages (E8 and E13) and two growth stages (D1 and D10). A total of 32,527 mRNAs were identified in pigeon skeletal muscles, including 14,378 novel mRNAs and 18,149 known mRNAs. A total of 2362 miRNAs were identified, including 1758 known miRNAs and 624 novel miRNAs. In total, 839 differentially expressed miRNAs (DEmiRNAs) and 11,311 differentially expressed mRNAs (DEGs) were identified. STEM clustering analysis assigned DEmiRNAs to 20 profiles, of which 7 were significantly enriched (*p*-value < 0.05). These seven significantly enriched profiles can be classified into two categories. The first category represents DEmiRNAs continuously downregulated from the developmental stage to the growth stage of pigeon skeletal muscle, and the second category represents DEmiRNAs with low expression at the development and early growth stage, and significant upregulation at the high growth stage. We then constructed an miRNA–mRNA network based on target relationships between DEmiRNAs and DEGs belonging to the seven significantly enriched profiles. Based on the connectivity degree, 20 hub miRNAs responsible for pigeon skeletal muscle development and growth were identified, including cli-miR-20b-5p, miR-130-y, cli-miR-106-5p, cli-miR-181b-5p, miR-1-z, cli-miR-1a-3p, miR-23-y, cli-miR-30d-5p, miR-1-y, etc. The hub miRNAs involved in the miRNA–mRNA regulatory networks and their expression patterns during the development and growth of pigeon skeletal muscle were visualized. GO and KEGG enrichment analysis found potential biological processes and pathways related to muscle growth and development. Our findings expand the knowledge of miRNA expression in pigeons and provide a database for further investigation of the miRNA–mRNA regulatory mechanism underlying pigeon skeletal muscle development and growth.

## 1. Introduction

Pigeon meat is rich in nutrition, high in protein, low in fat, and high in medicinal value. In China, pigeon meat is called “animal ginseng” and is considered an advanced nourishment product increasingly favored by consumers [1]. Meat production performance is an important index to measure the economic value of pigeons. However, the genetic improvement of pigeons’ meat production performance is relatively lagging compared with other poultry. The growth and development of skeletal muscle determine the meat production performance of pigeons [2]. As a result, understanding the molecular regulation mechanism of skeletal muscle growth and development is a crucial prerequisite for improving meat production performance through molecular selection technology [3].

The number of muscle cells and the development of skeletal muscle fibers are closely related to animal meat production. Myogenic regulatory factor family (MRF) members play essential roles in the process of skeletal muscle differentiation, including *MyoD* (myogenic regulatory factor 1), *MyoG* (myogenin), *MRF4* (muscle regulatory factor 4), and *Myf5* (myogenic factor 5) [4]. However, skeletal muscle growth and development involve multi-gene expression, signal transduction, and network regulation, and many regulatory factors remain to be identified.

MicroRNAs (miRNAs) are small, non-coding endogenous RNAs approximately 22 nt in length. miRNA regulates gene expression by binding 3′UTR of the target gene and then participates in various biological processes, including growth and development [5]. Increasing evidence shows that miRNA can act as a critical regulator to affect skeletal muscle development [4,6]. For example, miR-206 is involved in skeletal muscle development, growth/adaptation, regeneration, and muscle-related diseases [7]. Chen et al. found that miRNA-1 and miRNA-133 participated in regulating skeletal muscle proliferation and differentiation [8]. Kong et al. found that miR-17 and MIR-19 cooperate to promote skeletal muscle cell differentiation [9]. Cheng suggested that miR-223-3p promotes skeletal muscle regeneration by regulating inflammation in mice [10].

Although studies have shown that miRNA plays an essential role in the regulation of skeletal muscle growth and development, there is no report on the miRNA regulation of pigeon skeletal muscle growth and development. To explore the function of miRNA in pigeon skeletal muscle growth and development, this study will use next-generation sequencing technology to analyze the miRNA and mRNA expression in pigeon muscle growth and development, screen differentially expressed miRNA and mRNA, and construct miRNA–mRNA regulating networks based on expression data and the miRNA mRNA interaction theory. Our findings will contribute to a better understanding of the role of miRNA in pigeon muscle development and growth.

## 2. Materials and Methods

### 2.1. Animal Ethics Statement

The study was conducted following Chinese animal welfare guidelines, and the animal protocol was approved by the Animal Welfare Committee of Yangzhou University (permit number SYXK [Su] 2012–0029).

### 2.2. Preparation of Experimental Animals and Tissues

The pigeons used in this study were obtained from Wuxi Sanxiangan Agricultural Technology Development Co., Ltd. (Wuxi, China). The eggs were hatched by conventional procedures. The breast muscle tissues were collected at embryonic stage 8 (E8) and embryonic stage 13 (E13), and at one day (D1) and ten days (D10) post-hatch. All pigeons are euthanized, and each developmental period contains three biological replicates. All tissue samples were flash-frozen in liquid nitrogen and stored at −80°C.

### 2.3. RNA Sequencing

Total RNA was extracted using a Trizol reagent kit (Invitrogen, Carlsbad, CA, USA), according to the manufacturer’s protocol. RNA quality was assessed on an Agilent 2100 Bioanalyzer (Agilent Technologies, Palo Alto, CA, USA) and checked using RNase-free agarose gel electrophoresis. After the total RNA was extracted, rRNAs were removed to retain the mRNAs and ncRNAs. Then the enriched mRNA was fragmented into short pieces using fragmentation buffer and reverse transcripted into cDNA with random primers. Second-strand cDNA was synthesized by DNA polymerase I, RNase H, dNTP, and buffer. Then the cDNA fragments were purified, end-repaired, poly(A) was added, and then they were ligated to Illumina sequencing adapters with the QiaQuick PCR extraction kit (Qiagen, Venlo, The Netherlands). The ligation products were size-selected by agarose gel electrophoresis, PCR amplified, and sequenced at 10 × depth using the Illumina HiSeq^TM^ 2500, with 150 bp paired-end reads, from Gene Denovo Biotechnology Co., Ltd. (Guangzhou, China).

### 2.4. Small RNA Library Construction and Sequencing 

After the total RNA was extracted using the Trizol reagent kit (Invitrogen, Carlsbad, CA, USA), the RNA molecules in a size range of 18–30 nt were enriched by polyacrylamide gel electrophoresis (PAGE). Then the 3′ adapters were added, and the 36–44 nt RNAs were enriched. The 5′ adapters were then ligated to the RNAs as well. The ligation products were reverse transcribed by PCR amplification, and the 140–160 bp size PCR products were enriched to generate a cDNA library and sequenced at a 10 m depth using Illumina HiSeq^TM^ 2500, with 150 bp paired-end reads, from Gene Denovo Biotechnology Co., Ltd. (Guangzhou, China).

### 2.5. miRNA Identification and Quantification

Clean tags were obtained by removing the low-quality reads containing more than one low-quality (Q-value ≤ 20) base or containing unknown nucleotides (N), reads without 3′ adapters, reads containing 5′ adapters, reads containing 3′ and 5′ adapters, but without small RNA fragment between them, reads containing poly(A) in small RNA fragments, and reads shorter than 18 nt (not including adapters) with the Fast QC software (version 0.11.5, London, UK). Clean tags were aligned with the GenBank database (Release 209.0) [11] and Rfam database [12] to identify and remove rRNA, scRNA, snoRNA, snRNA, and tRNA. The clean tags were also aligned with the *Columba livia* reference genome. Tags mapped to exons or introns, which might be fragments from mRNA degradation, were removed. The tags mapped to repeat sequences were also removed. Then, the retained tags were aligned with the miRbase database (Release 21) to identify known miRNAs. All of the unannotated tags were aligned with the *Columba livia* reference genome. Novel miRNA candidates were identified according to their genome positions and hairpin structures, predicted by software Mireap_v0.2 (version 2.0, Redmond, WC, USA). The tags per million (TPM) algorithm was used to quantify the expression levels of the miRNAs.

### 2.6. mRNA Identification and Quantification

High-quality clean reads were obtained by removing reads containing adapters, reads consisting of all A bases, reads containing more than 10% of unknown nucleotides (N), and reads containing more than 50% of low quality (Q-value ≤ 20) bases using Fast QC software (version 0.11.5). rRNAs were removed by aligning high-quality clean reads with the ribosome RNA (rRNA) database. The rRNA-removed reads of each sample were then mapped to the *Columba livia* reference genome by TopHat2 (version 2.1.1), respectively. The fragments per kilobase of transcript per million mapped reads (FPKM) algorithm was used to quantify the expression levels of mRNAs.

### 2.7. Identification of Differentially Expressed Genes

To examine the differentially expressed mRNAs (DEMs) and miRNAs (DEmiRNAs) during the development and growth of pigeon skeletal muscle, six pairwise comparisons of the DEGs among the different groups were carried out, i.e., E8 vs. E13, E8 vs. D1, E8 vs. D10, E13 vs. D1, E13 vs. D10, and D1 vs. D10. The edgeR (version 3.14.0, Sydney, Australia) software was used to identify differentially expressed mRNAs (DEMs) and miRNAs (DEmiRNAs) with default parameters. These mRNAs and miRNAs with *p*-value < 0.05 and fold change ≥ 2 were considered as significantly differentially expressed.

### 2.8. miRNA Expression Pattern Clustering

Unsupervised hierarchical clustering and PCA (principal component analysis) were performed to explore the relationships between the samples using R packages. Clustering of DEmiRNAs was performed using the Short Time-series Expression Miner software (STEM) [13] on the OmicShare tools platform, a free online platform for data analysis (www.omicshare.com/tools, accessed on 1 March 2021). The parameters were set as follows: the maximum unit change in the model profiles between time points is 1, the maximum output profiles number is 20, and the minimum ratio of the fold change of DEmiRNAs is no less than 2.0. Profiles with a *p*-value < 0.05 were considered significant. 

### 2.9. Construction of miRNA–mRNA Network

First, we used RNAhybrid 2.1.2+svm_light 6.01 (Bielefeld, Germany), Miranda3.3a, (New York, NY, USA) and TargetScan 7.0 (Cambridge, MA, USA) software to predict the DEMis targets within the DEGs. miRNA sequences and family information were obtained from the TargetScan website (http://www.targetscan.org/, accessed on 1 March 2021). Then, the expression correlation between miRNA and the target was evaluated using the Pearson correlation coefficient (PCC). Pairs with PCC < −0.7 and *p* < 0.05 were selected as negatively co-expressed miRNA–mRNA pairs. The miRNA–mRNA network was visualized using Cytoscape software v 3.6.0 (Seattle, WA, USA) (http://www.cytoscape.org/, accessed on 28 March 2021). The connectivity degree of each node of the network was calculated. Hub miRNAs were identified by selecting the top 10 nodes with the largest connectivity degree. The hub miRNAs were visualized using Cytoscape v 3.6.0 and Origin 2021b (Northampton, MA, USA) software.

### 2.10. Functional Enrichment Analysis

To understand the underlying function of the miRNA–mRNA network, the Gene Ontology (GO) and the Kyoto Encyclopedia of Genes and Genomes (KEGG) pathway enrichment analyses were performed on the DEGs involved in the miRNA–mRNA network. GO or pathway terms with FDR < 0.05 were considered significant.

### 2.11. qRT-PCR Confirmation of Differentially Expressed Genes

MicroRNA was reverse-transcribed using the miRcute Plus miRNA First-Strand cDNA Kit (TIANGEN, Beijing, China) following the manufacturer’s recommendations. The miRNA forward primers were obtained commercially from Tsingke Biotechnology (Beijing, China). U6 snRNAs were simultaneously used as internal control genes (Table 1). ThemiRcute Plus miRNA qPCR Kit (TIANGEN, Beijing, China) was used to perform the quantitative real-time PCR (RT-qPCR) experiment, following the manufacturer’s instructions. All experiments were performed in triplicate. The selected gene expression was quantified using the comparative threshold cycle (2^−∆∆Ct^) method. The qRT-PCR results for all genes were statistically tested using the Student’s *t*-test.

## 3. Results

### 3.1. Analysis of miRNA and mRNA Expression in Pigeon Breast Muscle

A total of 32,527 mRNAs were identified in pigeon skeletal muscles, including 14,378 novel mRNAs and 18,149 known mRNAs. A total of 2362 miRNAs were identified, including 1758 known miRNAs and 624 novel miRNAs. Subsequently, we conducted hierarchical clustering and PCA analysis based on miRNA and mRNA expression across all 12 samples (Figure 1A,B). The results showed that the gene expression of pigeon skeletal muscle has a stage and time-specific pattern. Samples from embryonic stages (E8 and E13) were closely clustered into a subgroup, and samples from 1 and 10 days after hatching were clustered into a single group. The results indicated that the gene expression pattern of pigeon skeletal muscle in the embryonic stages was significantly different from that in the growth stages, suggesting that the genetic regulation mechanism in the two stages was different.

### 3.2. Identification of Differentially Expressed miRNA and mRNA

A total of 839 DEmiRNAs were identified among the four groups (*p*-value < 0.05, FC > 2) (Figure 1C). E8-D10 and E13-D10 comparisons contained the largest number of DEmiRNAs (E8-D10: 484; E13-D10: 461), while the D1-D10 comparison showed the smallest number of DEmiRNAs (174) (Data S1). A total of 11,311 DEGs were identified among the four groups (FDR < 0.05, FC > 2) (Figure 1D). E8-D10 and E8-D1 comparisons had the largest number of DEGs (E8-D10: 8242; E13-D10: 6024), while the E13-D1 comparison had the smallest number of DEGs (1959) (Data S1). These results indicate significant differences in the expression of miRNA and mRNA between the development and growth stages of pigeon skeletal muscle, suggesting that the potential mechanism of miRNA and mRNA regulating pigeon skeletal muscle development and growth is stage-specific.

### 3.3. miRNA Expression Profiles Clustering

The STEM algorithm was used to cluster the expression profile of DEmiRNAs in pigeon skeletal muscle growth and development [14]. The STEM clustering tool assigned each miRNA according to its time expression characteristics. A total of 20 expression profiles were generated, and 7 were significantly enriched (*p*-value < 0.05) (Figure 1E). The seven significantly enriched profiles can be classified into three categories. The first category includes four profiles, profile 0, profile 1, profile 2, and profile 7. There are 143, 34, 55, and 90 DEmiRNAs in these four profiles, respectively. The first category represents DEmiRNAs continuously downregulated from the developmental stage to the pigeon skeletal muscle growth stage. The second category includes profiles 10 and 19 and represents DEmiRNAs with low expression at the development and early growth stage and significantly upregulated expression at the high growth stage. Profiles 10 and 19 contain 57 and 36 DEmiRNAs, respectively. Profile 5, the third category, represents DEmiRNAs with a low expression at embryo stage 13, including 128 DEmiRNAs.

### 3.4. Target Gene Identification of DEmiRNAs in Significantly Enriched Profiles

In this study, we mainly focused on the dynamic expression of miRNA during the growth and development of pigeon skeletal muscle and explored miRNAs regulating its growth and development. Therefore, We take mainly the DEmiRNAs in the first and second categories of STEM cluster profiles as the research objects. The DEmiRNAs in the first category are highly expressed in the embryonic development stage, and the DEmiRNAs in the second category are highly expressed in the high growth period, indicating that these two categories of DEmiRNAs may play regulatory roles in the development and growth stages of pigeon skeletal muscle, respectively.

We predicted the target relationship between DEGs and DEmiRNAs in the first and second categories and analyzed their expression correlation. Based on target relationship prediction and expression correlation analysis (PCC < −0.7 and *p*-value < 0.05), 1289 target genes were identified for 322 DEmiRNAs in the first category, forming 13,569 miRNA–mRNA pairs (Appendix A). A total of 3310 target genes were identified for 93 DEmiRNAs in the second category, forming 20,149 miRNA–mRNA pairs (Appendix A).

### 3.5. Construction of the miRNA–mRNA Regulatory Network

To further explore key miRNAs that regulate the growth and development of pigeon skeletal muscle, we constructed the miRNA–mRNA regulatory network based on the results of the target relationship analysis of DEmiRNAs and DEGs in the two category profiles. miRNA–mRNA pairs with PCC < −0.9 were visualized by Cytoscape software. An miRNA–mRNA regulatory network was constructed, including 4045 miRNA–mRNA pairs composed of 195 DEmiRNAs and 1402 DEGs (Figure 2). 

### 3.6. Hub miRNA Identification

We used Cytoscape software to calculate the connectivity degree of nodes in the miRNA–mRNA network, and miRNAs with high connectivity degrees are considered to play important roles in the network. In this study, the top 10 miRNAs according to degree values were considered the hub genes. The hub miRNAs of the first category profiles are cli-miR-20b-5p, miR-130-y, cli-miR-106-5p, cli-miR-181b-5p, cli-miR-456-3p, cli-miR-1677-3p, cli-miR-1677-5p, cli-miR-130c-5p, cli-miR-103-5p, and cli-miR-18a-5p. The hub miRNAs of the second category profiles are miR-27-y, miR-1-z, cli-miR-1a-3p, miR-23-y, cli-miR-30d-5p, miR-1-y, miR-133-y, miR-26-x, cli-miR-30c-5p, and miR-101-y (Table 2). The subnetwork containing hub miRNAs and their target mRNAs was then visualized using Cytoscape software (Figure 3A and Figure 4A). The expression patterns of these hub miRNAs during the development and growth of pigeon skeletal muscle were also analyzed. The results showed that the average degree of hub miRNAs in Network A was 44, and cli-miR-20b-5p had the highest connectivity degree (60). All the ten hub miRNAs were highly expressed during embryonic development and significantly downregulated during the growth stage of pigeon skeletal muscle (Figure 3B). The average degree of hub miRNAs in Network B was 192, and miR-27-y had the highest degree (259). All the ten hub miRNAs were significantly upregulated during the skeletal muscle growth stage (Figure 4B).

### 3.7. Functional Enrichment Analysis of Target Genes

To further explore the possible molecular mechanism by which miRNA regulates the growth and development of pigeon skeletal muscle, we performed GO and KEGG enrichment analyses on target mRNAs of the miRNAs in Network A and network B (Figure 5). GO enrichment analysis showed that target genes of miRNAs in Network A are significantly enriched in 47 biological processes (FDR < 0.05), including biological processes such as striated muscle cell differentiation, striated muscle cell development, and muscle cell development. Target genes of miRNAs in Network B were significantly enriched in 242 biological processes (FDR < 0.05), including the regulation of biological processes, the regulation of cellular processes, and tissue morphogenesis biological processes. KEGG pathway enrichment showed that target genes of miRNAs in Network A are significantly enriched in 29 signal pathways (FDR < 0.05), including signal pathways such as ECM-receptor interaction, focal adhesion, and the biosynthesis of amino acids. Target genes of miRNAs in Network B were significantly enriched in four signal pathways (FDR < 0.05), including signal pathways such as cell growth and death, signal transduction, and development.

### 3.8. qRT-PCR

To confirm the differentially expressed genes between the embryonic and growth groups obtained by RNA-seq, four differentially expressed genes (miR-130-y, miR-27-y, miR-26-x, and cli-miR-30c-5p) in the network were selected, and their expression patterns were quantified by qRT-PCR. The expression of the above miRNAs during the growth period is higher than the expression during the embryonic stage. The results were in line with our sequencing results, highlighting the reliability of our sequencing data (Figure 6).

## 4. Discussion

Meat performance is one of the important indicators to measure the economic value of pigeons. The growth and development of skeletal muscle determine the meat performance of pigeons [15]. Elucidating the molecular regulatory mechanism of pigeon skeletal muscle growth and development is a crucial prerequisite for improving the production performance of pigeon meat using molecular breeding technology. However, compared with other poultry, the molecular mechanism of the genetic regulation of pigeon skeletal muscle growth and development remains unclear, especially in miRNA regulation. In the present study, to screen miRNAs that regulate pigeon skeletal muscle growth and development and to explore its possible molecular mechanism, next-generation sequencing technology was used to characterize the dynamic expression of miRNA and mRNA during the development and growth of pigeon skeletal muscle. This study presents the first integrated analysis of miRNA and mRNA expression in pigeon skeletal muscle. Hierarchical cluster analysis was performed on samples from different developmental periods based on the expression of miRNA and mRNA. The results showed that the samples from the embryonic stage (E8 and E13) were clustered into a subgroup, and there were significant differences in gene expression between the samples from the embryonic and growth stages. PCA analysis further showed a close proximity between samples from the embryonic stage and a greater distance between samples from the embryonic stage and the growth stage. These results suggest different regulation mechanisms of pigeon skeletal muscle between the growth and embryonic development stages.

The STEM algorithm is widely used to cluster expression data sampled in a time sequence and analyze its expression pattern [13,14]. We performed cluster analysis on DEmiRNAs using the STEM algorithm. Two categories of significantly enriched profiles were identified. The first category included 322 DEmiRNAs, which were highly expressed during embryonic development and significantly downregulated during the growth stage. This suggests that these DEmiRNAs may play critical regulatory roles in pigeon skeletal muscle embryonic development. The second category included 93 DEmiRNAs, which were lowly expressed during the embryonic stage and significantly upregulated during the growth stage, suggesting that these DEmiRNAs may play important regulatory roles in pigeon skeletal muscle growth.

We constructed an miRNA–mRNA regulatory network based on expression data and the miRNA-mRNA interaction theory. It was found that an miRNA can target multiple mRNAs, and multiple miRNAs can also regulate one mRNA. The regulatory relationship between the differentially expressed miRNA and mRNA which we discovered will contribute to a better understanding of the role of miRNAs in pigeon muscle development and growth.

To further screen candidate miRNAs that regulate pigeon skeletal muscle growth and development, we constructed two miRNA–mRNA regulatory networks based on the expression correlation and target relationship prediction analysis of DEmiRNAs and DEGs. Network A included DEmiRNAs in the first category of profiles and Network B included DEmiRNAs in the second category of profiles. According to the connectivity degree, 20 candidate miRNAs were identified, including miR-130-y, cli-miR-106-5p, cli-miR-181b-5p, cli-miR-456-3p, miR-1-z, cli-miR-1a-3p, miR-23-y, cli-miR-30d-5p, etc. Many miRNAs are known to be involved in the regulation of muscle growth and development. Luo found that miR-20b-5p promotes muscle cell differentiation and inhibits muscle cell proliferation by directly binding to the 3′UTR of E2F transcription factor 1 (*E2F1*) mRNA [16]. Zhao et al. found that miR-181b-5p can promote skeletal muscle growth by reducing the *MSTNb* protein level of tilapia [17]. Guanidinoacetic acid (*GAA*) supplements can promote muscle cell differentiation and skeletal muscle growth by activating the AKT/mTOR/S6K signaling pathway induced by miR-133a-3p and miR-1a-3p [18]. miR-30c-5p is related to the treatment of skeletal fibroids with myocardial injury [19]. Postmenopausal osteoporosis (PMOP) is a metabolic bone disease caused by imbalance between osteoblast bone formation and osteoclast bone resorption, and *DGCR5* up-regulates Runx2 through miR-30d-5p to induce osteogenic differentiation, which contributes to ameliorate postmenopausal osteoporosis (PMOP), further affecting the growth of skeletal muscle [20]. miR-18a-5p has also been confirmed to promote smooth muscle cell proliferation and migration by activating the AKT/extracellular signal-regulated protein kinases (ERK) signaling pathway [21]. The above miRNA were identified as hub miRNAs in our study, suggesting their potential role in the growth and development of pigeon skeletal muscle. In addition, the functions of cli-miR-130-y, cli-miR-1677-3p, cli-miR-1677-5p, cli-miR-130c-5p, cli-miR-103-5p, miR-27-y, miR-1-z, miR-23-y, cli- miR-1-y, miR-133-y, miR-26-x, and miR-101-y in muscle growth and development have not been reported.

To explore the potential mechanism of miRNA regulating the growth and development of pigeon skeletal muscle, we performed GO and KEGG enrichment analyses of mRNA in Network A and Network B, respectively. The GO enrichment analysis showed that the target genes in Network A were significantly enriched in biological processes related to skeletal muscle cell differentiation and development, such as striated muscle cell differentiation, striated muscle cell development, and muscle cell development. This indicates that miRNAs in Network A may play important regulatory roles in the differentiation and development of pigeon skeletal muscle cells. The target genes in Network B are significantly enriched in 242 biological processes, such as regulation of biological process, regulation of cellular process, and tissue morphogenesis, indicating that the 242 biological processes may be related to the growth of pigeon skeletal muscle. However, the exact relationship between these biological processes and pigeon muscle development still requires further investigation. We also found that target genes in Network A and Network B were enriched in different biological processes, which indicates different mechanisms by which miRNAs regulate the embryonic development and growth of pigeon skeletal muscle.

KEGG is a database for the systematic analysis of genetic function and genomic information which can be used as a complete network for studying information about genes and expression. [22]. Pigeon muscle growth is a complex process influenced by multiple genes and controlled by multiple pathways. Many of the pathways enriched by mRNA in Network A are related to muscle growth and development. For example, mRNAs in Network A were enriched in ECM receptor interaction, focal adhesion, and the biosynthesis of amino acids pathways. ECM-receptor interaction and focal adhesion pathways are frequently found in different developmental stages of chicken breast muscle [23,24]; the content and synthesis of amino acids also affect the development of skeletal muscle [25]. Focal adhesions are large-scale protein complexes on the surface of cell substrates. They physically connect the extracellular matrix with the cytoskeleton. They have long been speculated to mediate cell migration [26]. Proximal multiplier activated receptors (PPARs) regulate genes involved in development, metabolism, inflammation, and many cellular processes in different organs. PPAR is also present in muscles and exerts a pectoral muscle-specific response when activated by its ligand [27]. Ca2 + itself or Ca2 + dependent signaling pathways play a key role in various cellular processes, from cell growth to death. The most typical example is that of skeletal muscle cells [28].

Similarly, mRNAs in Network B are also enriched in pathways related to muscle growth and development. For example, muscle growth is regulated by signal transduction pathways that sense and compute local and systemic signals and regulate various cellular functions [29]. The Hippo signaling pathway is an important mediator for the tissue growth of some epithelial cell types. A study has shown that the Hippo signaling pathway serves as a key element in muscle fibers and muscle satellite cells [30] and is required for early muscle development [31]. The biogenesis of ribosomes in eukaryotes has become a major regulator of the growth and maintenance of the skeleton. [32]. Our findings suggest the potential roles of the above pathways in regulating pigeon skeletal muscle development and growth. However, further study is still required. Moreover, target mRNAs in Network A and Network B are significantly enriched in different pathways, which again indicates differences in the potential mechanisms of miRNA to regulate skeletal muscle growth and embryonic development.

## 5. Conclusions

In the present study, a total of 839 DEmiRNAs and 11,311 DEGs were identified in 12 pigeon skeletal muscle samples. STEM clustering analysis assigned DEmiRNAs to 20 profiles, of which 7 were significantly enriched. An miRNA–mRNA network was constructed based on target relationships between DEmiRNAs and DEGs belonging to the seven significantly enriched profiles. Based on the connectivity degree, 20 hub miRNA responsible for pigeon skeletal muscle development and growth were identified, including cli-miR-20b-5p, miR-130-y, cli-miR-106-5p, cli-miR-181b-5p, miR-1-z, cli-miR-1a-3p, miR-23-y, cli-miR-30d-5p, miR-1-y, etc. These hub miRNAs may regulate pigeon skeletal muscle development and growth by regulating target genes.

## Figures and Tables

**Figure 1 animals-12-02509-f001:**
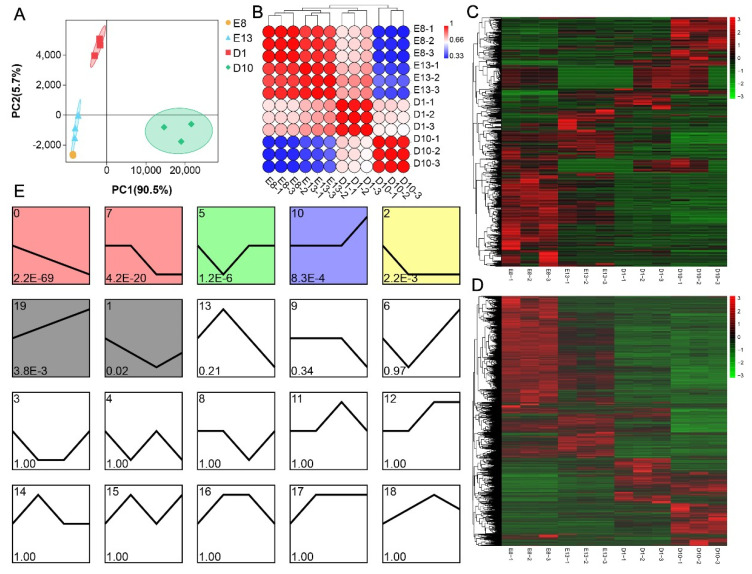
Expression patterns of miRNAs and mRNAs during pigeon skeletal muscle development and growth. (**A**) Principal component analysis (PCA) of the samples based on all miRNA and mRNA expressions. (**B**) Correlation analysis of the samples based on all miRNA and mRNA expressions. The colors from blue to red represent a correlation from low to high. The principal component and correlation analysis indicate distinct expression patterns of miRNA and mRNA at different stages. There were differences among the regulatory mechanisms for the development and growth of skeletal muscle. (**C**) Heatmap of differentially expressed miRNA. (**D**) Heatmap of differentially expressed mRNA. Colors from green to red indicate expression levels from low to high. (**E**) Expression profiles analysis of all miRNA by Short Time-series Expression Miner (STEM) program. A total of 20 profiles were clustered. Colored profiles were significant (*p*-value < 0.05).

**Figure 2 animals-12-02509-f002:**
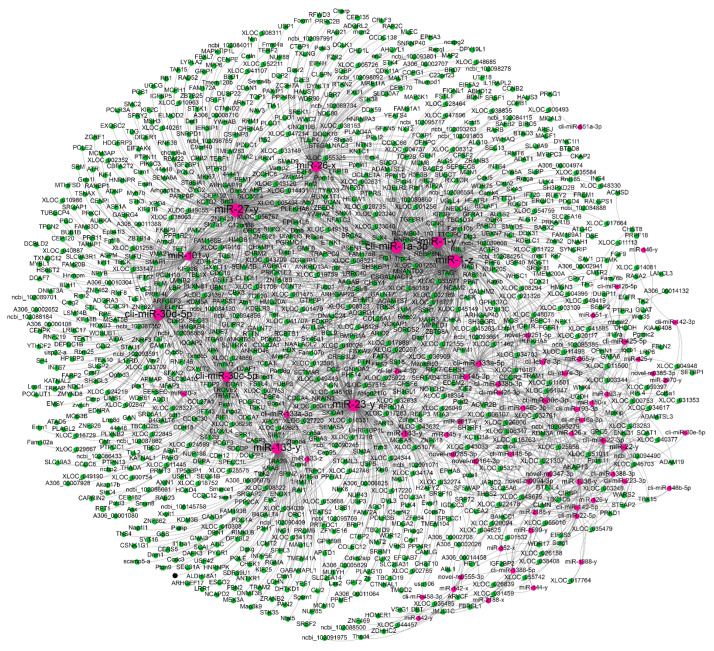
Visualization of the miRNA–mRNA regulatory networks.

**Figure 3 animals-12-02509-f003:**
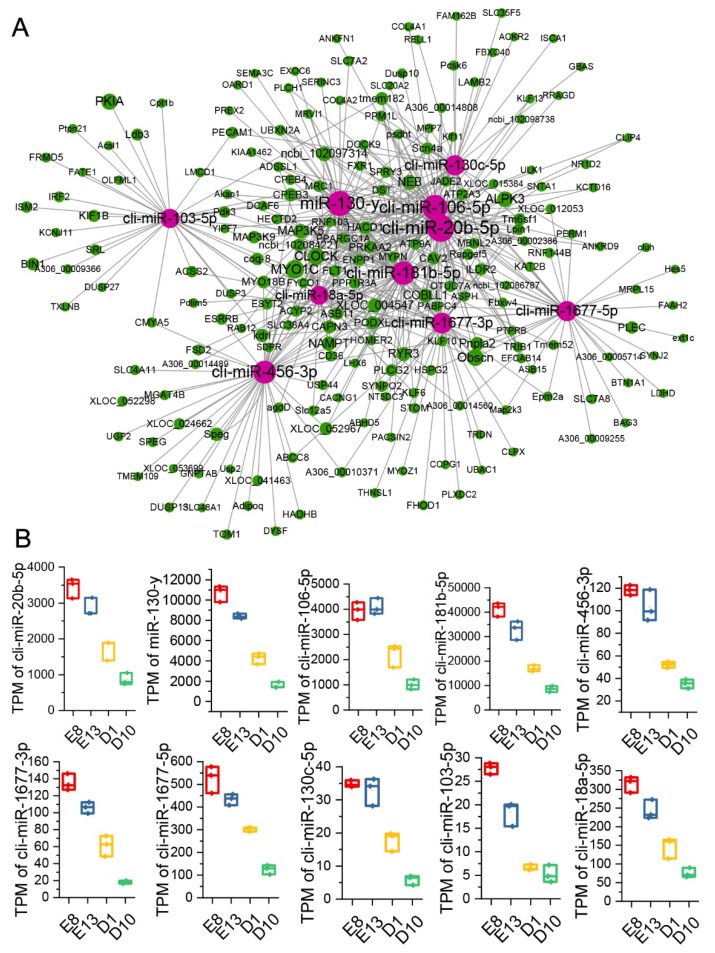
Identification and visualization of hub miRNAs in Network A. (**A**) Visualization of interactions between hub miRNAs and their target genes in Network A. The red node represents miRNA, the green node represents mRNA, and the size of the node represents the connectivity degree, respectively. (**B**) The expression patterns of the ten hub miRNAs during the growth and development of pigeon skeletal muscle. All the ten hub miRNAs are highly expressed during the embryonic development stage, and their expression is significantly downregulated during the growth stage.

**Figure 4 animals-12-02509-f004:**
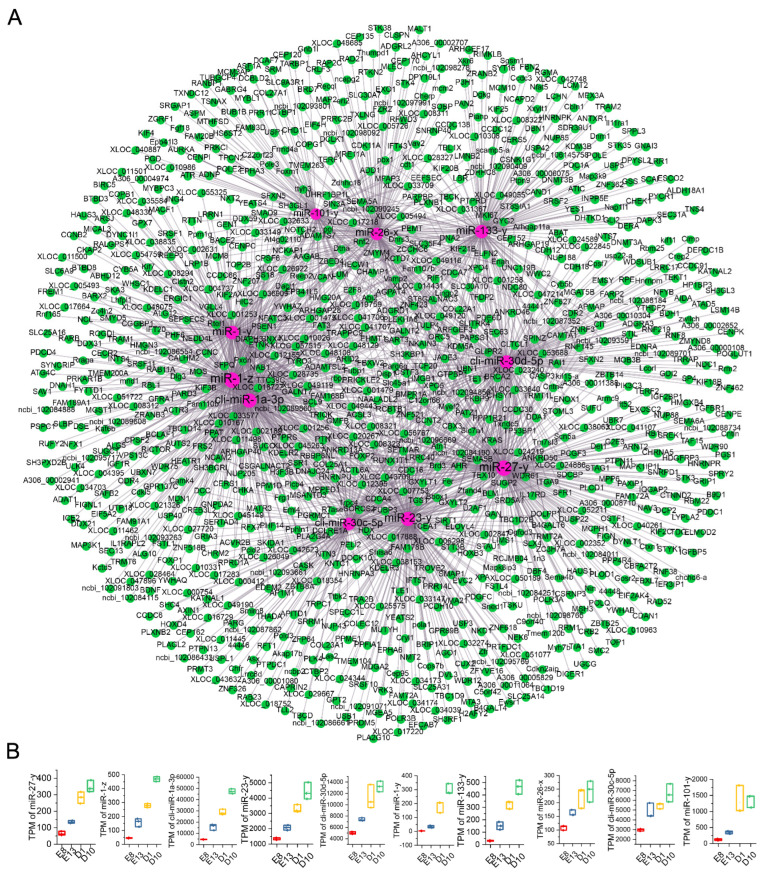
Identification and visualization of hub miRNAs in Network B. (**A**) Visualization of interactions between hub miRNAs and their target genes in Network B. The red node represents miRNA, the green node represents mRNA, and the node’s size represents degree, respectively. (**B**) The expression patterns of the ten hub miRNAs during the growth and development of pigeon skeletal muscle. The expression of all ten hub miRNAs was significantly upregulated during the growth stage.

**Figure 5 animals-12-02509-f005:**
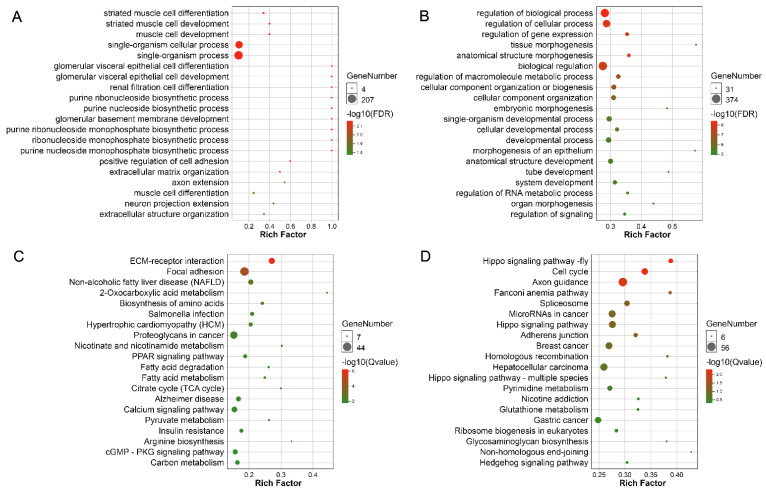
GO and KEGG enrichment of target genes. (**A**) The GO enrichment analysis of target genes of miRNAs in the first category of profiles. (**B**) The GO enrichment analysis of target genes of miRNAs in the second category of profiles. (**C**) The KEGG pathway enrichment analysis of target genes of miRNAs in the first category of profiles. (**D**) The KEGG pathway enrichment analysis of target genes of miRNAs in the second category of profiles.

**Figure 6 animals-12-02509-f006:**
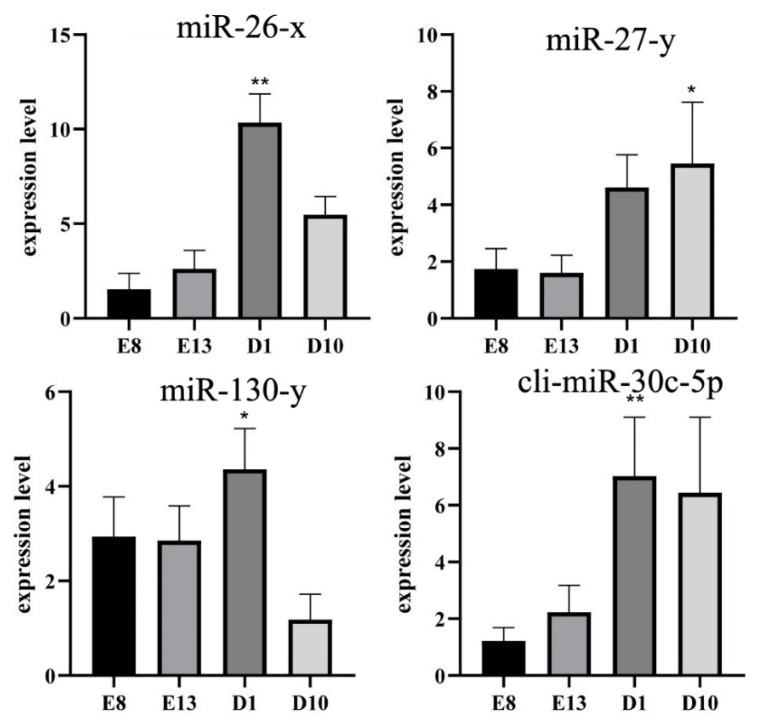
Validation of hub genes using RT-qPCR. ** indicates a very significant difference (*p* < 0.01); * indicates a significant difference (*p* < 0.05).

**Table 1 animals-12-02509-t001:** Primers for RT-qPCR.

Genes	Primer Sequences
miR-26-x	CAAGTAATCCAGGATAGGCT
miR-27-y	TTCACAGTGGCTAAGTTCC
cli-miR-30c-5p	GTAAACATCCTACACTCTCAG
miR-130-y	CAGTGCAATAATGAAAGGG
U6	GATGACACGCAAATTCGTGAAG

**Table 2 animals-12-02509-t002:** Pigeon skeletal muscle growth and development related hub miRNAs.

Network	Hub miRNA	Degree
Network A	cli-miR-20b-5p	60
	miR-130-y	53
	cli-miR-106-5p	53
	cli-miR-181b-5p	48
	cli-miR-456-3p	46
	cli-miR-1677-3p	39
	cli-miR-1677-5p	37
	cli-miR-130c-5p	37
	cli-miR-103-5p	34
	cli-miR-18a-5p	33
Network B	miR-27-y	259
	miR-1-z	254
	cli-miR-1a-3p	217
	miR-23-y	210
	cli-miR-30d-5p	192
	miR-1-y	191
	miR-133-y	184
	miR-26-x	154
	cli-miR-30c-5p	150
	miR-101-y	110

## Data Availability

The data supporting this study’s findings are available in Genome Sequence Archive (https://ngdc.cncb.ac.cn/gsa/ (accessed on 16 September 2020)), with reference Numbers CRA005062 and CRA005074.

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
