# Peer review of "Identification of miRNA–mRNA Networks Associated with Pigeon Skeletal Muscle Development and Growth"

_animals, 2022, doi:10.3390/ani12192509_

Round 1
Reviewer 1 Report
Summary:
This manuscript aims to identify miRNA and RNA networks that are influencing muscle growth in pigeons kept for meat production. The experiment looked at multiple time points pre- and post-hatching; and used NGS techniques, differential expression analysis, STEM clustering, and functional enrichment analysis to investigate the research question. The analyses found novel miRNAs that are linked to numerous RNAs that influence muscle function and growth.
General comments:
- The line numbering is missing from the PDF makes it difficult to review or refer to sentences. Efforts were made to refer using section numbers.
- Could you please make sure you provide references for the claims on "Pigeon meat, medicinal value etc" in the first paragraph of your introduction.
- Use cases of finding novel functional annotations of pigeon genome i.e miRNAs and their targets, requires a bit more expansion in the introduction. Is the aim to help breeding programmes, disease biomarkers ? or is there other economical/cultural contexts? Recent reviews have described such approaches in detail: https://doi.org/10.3390%2Fijms22063080
- As majority of the novel miRNAs were detected based on incompleteness of the pigeon genome (remapping unmapped reads to Gallus gallus), its vital that the important of the DEmiRNAs are not over stated in discussion as it currently is. Please incorporate this point in your first paragraph of the discussion.
- More granular details are needed for the bioinformatics tools used in the study in order to make this dataset F.A.I.R.E.R compliant. Please be exhaustive in mentioning the tool version, potential scripts (supplement them) and code repositories.
- Its highly recommended to make the genomic data available if there is no commercial conflict of interest or background IP
Specific comments:
- Include euthanasia protocol if possible in section 2.2
- Please include length and depth of sequnecing in 2.3 and 2.4
- 2.3 why there is eukaryote vs prokaryote RNA separation? please clarify the reason for this methodology and expand the text a bit. Its not clear if the RNA isolation was done in parallel for each sample or not.
- 2.5 include all the tools and the versions used.
- 2.5 include the reference genome accession number and source (Gallus gallus)
- 2.6 again please include reference genome accession number and source (Columba livia)
- 2.7 FDR correction was carried out on the p values or is this raw p value 0.05
- Figures 2B,4A are almost unreadable. Please make sure you maximise the font size and reduce granularity for better visuals
- Consider matching layout for figure 4 to that of figure 3. The matrix of boxplots in figure 3 is more legible as they have more space under the network diagram
- Introduction - Please add references concerning the nutritive quality of pigeon meat and consumer uptake
- Methods 2.5 - How can you be sure that the miRNA tags aligned to the gallus gallus genome are true miRNA hits, and not just aligning with their target mRNA?
- 2.5 Why did you choose the gallus gallus genome for this alignment?
- After aligning novel miRNAs to the chicken genome, has this allowed you to improve the pigeon annotation with this information?
- 2.5/6 Please can you detail how you cleaned/trimmed the reads? What package was used for this? include the trimming flags if possible
- 2.7 There is very little information on the use of EdgeR, can you add some detail to make it clear what settings were used. For example, what (if any) filtering was carried out?
- 2.7 When carrying out the DE analysis with EdgeR, what design(s) was used? In results 3.2 you discuss which comparisons had the most and least DEmi/mRNAs, it would be clearer if you include which comparisons were carried out in this part of the methods as well.
- Discussion page 13 (GO & KEGG analysis). The pathways 'biological process, regulation of cellular process, and tissue morphogenesis' have been highlighted for network B. Would you expect these high-level pathways to be enriched in other tissues? Were there any other significant pathways that better illustrate a link to muscle growth?
Author Response
Reviewer 1:
General comments:
- Could you please make sure you provide references for the claims on "Pigeon meat, medicinal value etc" in the first paragraph of your introduction.
Response: The reference has been added to the manuscript.
- Use cases of finding novel functional annotations of pigeon genome i.e miRNAs and their targets, requires a bit more expansion in the introduction. Is the aim to help breeding programmes, disease biomarkers ? or is there other economical/cultural contexts? Recent reviews have described such approaches in detail: https://doi.org/10.3390%2Fijms22063080
Response: Since the molecular regulation mechanism of miRNA is rarely studied in pigeons, our research is to find the regulatory relationship between miRNA and mRNA, lay the foundation for the growth and development of pigeon skeletal muscle regulation mechanism, and provide a theoretical basis for breeding plans.
- As majority of the novel miRNAs were detected based on incompleteness of the pigeon genome (remapping unmapped reads to Gallus gallus), its vital that the important of the DEmiRNAs are not over stated in discussion as it currently is. Please incorporate this point in your first paragraph of the discussion.
Response: The use of edgeR software to screen for DEmiRNAs in pigeon muscles suggests that these DEmiRNAs may have a regulatory effect on the growth and development of pigeon skeletal muscle, but the specific regulatory mechanism needs to be further verified.
- More granular details are needed for the bioinformatics tools used in the study in order to make this dataset F.A.I.R.E.R compliant. Please be exhaustive in mentioning the tool version, potential scripts (supplement them) and code repositories.
Response: Tool versions and possible scripts have been supplemented in the appropriate places in the article
- Its highly recommended to make the genomic data available if there is no commercial conflict of interest or background IP
Response: The data that support the findings of this study are available in Genome Sequence Archive (https://ngdc.cncb.ac.cn/gsa/ accessed on 16 September 2020) with reference Numbers CRA005062 and CRA005074.
Specific comments:
- Include euthanasia protocol if possible in section 2.2
Response: “All animals are euthanized” has been added to 2.2.
- Please include length and depth of sequnecing in 2.3 and 2.4
Response: The length and depth of sequnecing have been added in 2.3 and 2.4. The mRNA libraries were sequenced at 10x depth using Illumina HiSeqTM 2500 with 150 bp paired-end reads. The miRNA libraries were sequenced at 10M depth using Illumina HiSeqTM 2500 with 150 bp paired-end reads.
- 2.3 why there is eukaryote vs prokaryote RNA separation? please clarify the reason for this methodology and expand the text a bit. Its not clear if the RNA isolation was done in parallel for each sample or not.
Response: This is an expression mistake. We are so sorry for this mistake. It has been corrected to “After total RNA was extracted, rRNAs were removed to retain mRNAs and ncRNAs.”
- 2.5 include all the tools and the versions used.
Response: The version number has been added in 2.5
- 2.5 include the reference genome accession number and source (Gallus gallus)
Response: The "Gallus gallus" has been changed to "Columba livia". The corresponding sentence has been revised to " All of the unannotated tags were aligned with the Columba livia reference genome.”
- 2.7 FDR correction was carried out on the p values or is this raw p value 0.05
- Figures 2B,4A are almost unreadable. Please make sure you maximise the font size and reduce granularity for better visuals
Response: Figure 2 has been corrected in the manuscript
- Consider matching layout for figure 4 to that of figure 3. The matrix of boxplots in figure 3 is more legible as they have more space under the network diagram
Response: The layouts in Figure 3 and Figure 4 have been changed to match the manuscript
- Introduction - Please add references concerning the nutritive quality of pigeon meat and consumer uptake
Response: References have been added to the manuscript.
- Methods 2.5 - How can you be sure that the miRNA tags aligned to the Gallus gallus genome are true miRNA hits, and not just aligning with their target mRNA?
- 2.5 Why did you choose the gallus gallus genome for this alignment?
Response: Thanks for your comment. The "Gallus gallus" has been changed to "Columba livia" in the manuscript.
- After aligning novel miRNAs to the chicken genome, has this allowed you to improve the pigeon annotation with this information?
Response: Has been changed to After aligning novel miRNAs to the pigeon genome.
- 2.5/6 Please can you detail how you cleaned/trimmed the reads? What package was used for this? include the trimming flags if possible
Response: The package and methods that cleaned the reads have been added. For miRNA, the clean tags were obtained by removing low-quality reads containing more than one low-quality (Q-value ≤ 20) base or containing unknown nucleotides(N), removing reads without 3' adapters, removing reads containing 5' adapters, removing reads containing 3' and 5' adapters but without small RNA fragment between them, removing reads con-tain-ing ployA in small RNA fragment, and removing reads shorter than 18nt (not include adapters) with the Fast QC software (version 0.11.5). For mRNA, the high-quality clean reads were obtained by removing reads containing adapters, reads consisting of all A bases, reads containing more than 10% of unknown nucleotides (N), and reads containing more than 50% of low quality (Q-value ≤ 20) bases with Fast QC software (version 0.11.5).
- 2.7 There is very little information on the use of EdgeR, can you add some detail to make it clear what settings were used. For example, what (if any) filtering was carried out?
Response: In the present study, the edgeR (version 3.14.0) software was used to identify differentially expressed mRNAs (DEMs) and miRNAs (DEmiRNAs) with default parameters. mRNAs and miRNAs with P-value < 0.05 and fold change ≥ 2 were considered as DEMs and DEmiRNAs. This has been added to section 2.7.
- 2.7 When carrying out the DE analysis with EdgeR, what design(s) was used? In results 3.2 you discuss which comparisons had the most and least DEmi/mRNAs, it would be clearer if you include which comparisons were carried out in this part of the methods as well.
Response: To examine the differentially expressed mRNAs (DEMs) and miRNAs (DEmiRNAs) during the development and growth of pigeon skeletal muscle, six pairwise comparisons of the DEGs among the different groups were carried out, i.e., E8 v.s. E13, E8 v.s. D1. E8 v.s. D10, E13 v.s. D1, E13 v.s. D10, and D1 v.s. D10. The above information has been added to section 2.7.
- Discussion page 13 (GO & KEGG analysis). The pathways 'biological process, regulation of cellular process, and tissue morphogenesis' have been highlighted for network B. Would you expect these high-level pathways to be enriched in other tissues? Were there any other significant pathways that better illustrate a link to muscle growth?
Response: In fact, we found that target genes of miRNAs in network B were significantly enriched in 242 biological processes (FDR < 0.05). Regulation of biological process, regulation of cellular process and tissue morphogenesis biological processes are just three of the significantly enriched pathways. We expect that these high-level pathways might be enriched in other tissues. Thus, in the present study, we just speculate that some of the 242 significantly enriched biological processes may be related to pigeon skeletal muscle development. The exact relationship between these biological processes and pigeon muscle development still requires further investigation.

Reviewer 2 Report
Manuscript entitled "Identification of miRNA-mRNA Networks Associated with Pigeon Skeletal Muscle Development and Growth" may be accepted after incorporated minor corrections highlighted in the manuscript.

Author Response
Reviewer 2:
1 “MiRNA” replaced with “miRNA”
Response: “MiRNA” has been changed to “miRNA”.
2 Please,specify the number of samples to be taken in present study
Response: We analyzed four periods, three biological replicates in each period, for a total of 12 samples.
3 Q-value replaced with q-value
Response: “Q-value” has been changed to “q-value”.
4 The labels in Figure 1 (part C and D) are difficult to read, please improve it
Response: The clarity of Figure 1 has been adjusted in the manuscript
5 The labels in Figure 2 are difficult to read, please improve it
Response: The clarity of Figure 2 has been adjusted in the manuscript
6 The labels in Figure 4 (part A) are difficult to read, please improve it
Response: The clarity of Figure 4 has been adjusted in the manuscript

Reviewer 3 Report
Page 1, paragraph 1: change to "The growth and development of skeletal muscle determines...."
Page 1, paragraph 2: MyoD is "myogenic determination factor 1", edit the word regulatory.
Results, section 3.6: Authors need to better explain the importance of connectivity degree.
Figure 5: Labels are not readable when the article is printed.
Figure 6: Put miRNA titles at the top of the figure centered and bolded. Font should be larger than x and y axis.
Figure 6 legend is lackluster. Authors need to better describe how RT-qPCR validates their prior data in the figure legend.
Statistics are missing from each figure legend. These must be included.
In the discussion, there is no clear relation back to the purpose of the study. The discussion must include a section that links the purpose with the results.
Author Response
Reviewer 3:
Comments and Suggestions for Authors
Page 1, paragraph 1: change to "The growth and development of skeletal muscle determines...."
Response: Changed to "The growth and development of skeletal muscle determines...."
Page 1, paragraph 2: MyoD is "myogenic determination factor 1", edit the word regulatory.
Response: Changed to "myogenic regulatory factor 1"
Results, section 3.6: Authors need to better explain the importance of connectivity degree.
Response: It has already been mentioned in the third paragraph of the introduction “miRNA regulates gene expression by binding 3’UTR of the target gene, and then participates in a variety of biological processes, including growth and development.”
Figure 5: Labels are not readable when the article is printed.
Response: Figure 5 has been corrected in the manuscript.
Figure 6: Put miRNA titles at the top of the figure centered and bolded. Font should be larger than x and y axis.
Response: Figure 6 has been corrected in the manuscript.
Figure 6 legend is lackluster. Authors need to better describe how RT-qPCR validates their prior data in the figure legend.
Response: Thanks for your comment. “The expression during the growth period is higher than the expression during the embryonic stage” has been added to the manuscript.
Statistics are missing from each figure legend. These must be included.
Response: This has been added Note: **indicates a very significant difference (P<0.01), * indicates a significant difference (P<0.05).
In the discussion, there is no clear relation back to the purpose of the study. The discussion must include a section that links the purpose with the results.
Response: This has been added “We construct a miRNA-mRNA regulatory network based on expression data and miRNA mRNA interaction theory. It was found that a miRNA can target multiple mRNAs, and one mRNA can also be regulated by multiple miRNAs. The regulatory relationship between the differentially expressed miRNA and mRNA we found will contribute to a better understanding of the role of miRNAs in pigeon muscle development and growth”in the discussion.
